

# Predicting early-onset COPD risk in adults aged 20–50 using electronic health records and machine learning

Guanglei Liu[1], Jiani Hu[2], Jianzhe Yang[2] and Jie Song[2]

[1] School of Information Science and Engineering, Yunnan University, Kunming, Yunnan, China
[2] Ailurus Biotechnology Ltd., Shenzhen, Guangdong, China

## ABSTRACT

Chronic obstructive pulmonary disease (COPD) is a major public health concern, affecting estimated 164 million people worldwide. Early detection and intervention strategies are essential to reduce the burden of COPD, but current screening approaches are limited in their ability to accurately predict risk. Machine learning (ML) models offer promise for improved accuracy of COPD risk prediction by combining genetic and electronic medical record data. In this study, we developed and evaluated eight ML models for primary screening of COPD utilizing routine screening data, polygenic risk scores (PRS), additional clinical data, or a combination of all three. To assess our models, we conducted a retrospective analysis of approximately 329,396 patients in the UK Biobank database. Incorporating personal information and blood biochemical test results significantly improved the model's accuracy for predicting COPD risk, achieving a best performance of 0.8505 AUC, a specificity of 0.8539 and a sensitivity of 0.7584. These results indicate that ML models can be effectively utilized for accurate prediction of COPD risk in individuals aged 20 to 50 years, providing a valuable tool for early detection and intervention.

## INTRODUCTION

Chronic obstructive pulmonary disease (COPD) is a serious and complex disease that affects an estimated 164 million people around the world. It is characterized by a progressive decline in lung function, leading to breathlessness and limitations in physical activity. COPD is primarily caused by smoking, although air pollution, occupational exposure, and genetics can also be contributing factors. Early diagnosis of COPD could facilitate preventive measures, such as smoking cessation or adjustment of environmental exposures, and allow proper management of the disease (*Singh et al., 2019*).

Currently, screening programs exist for at-risk populations, such as those older than 45 years old with a history of smoking, and in most cases rely on self-reported smoking history, lung function tests, and imaging scans. For example, in a study by *Raju et al. (2018)*, elderly patients over 65 years of age were monitored through annual screening programs to determine the risk of developing COPD. Although this type of screening

Corresponding author
Jie Song, avec@ailurus.bio

program is helpful, it does not account for the role of genetics and other patient data points, and only identifies those at risk who are already symptomatic (*Singh et al., 2019*).

In recent years, machine learning (ML) models have emerged as a powerful tool for identifying disease risks, including chronic obstructive pulmonary disease (COPD) (*Ma et al., 2020*; *Peng et al., 2020*; *Zhang et al., 2022a*). Extensive research has demonstrated the accuracy of ML models in predicting the development of COPD based on genetic information and electronic medical records data (*Peng et al., 2020*; *Zhang et al., 2022a*). In particular, studies by *Peng et al. (2020)* and *Cosentino et al. (2023)* showcased the effectiveness of ML algorithms in detecting early signs of COPD from chest CT scans and in predicting the case-control status of COPD from raw high-dimensional spirograms, respectively (*Sugimori et al., 2021*). Makimoto's study, focusing on CT-based quantitative models to predict severe exacerbations of COPD (*Makimoto & Kirby, 2023*).

The application of polygenic risk scores (PRSs) in evaluating the risk of COPD has attracted significant attention (*Zhang et al., 2022b*, *2022c*). *Zhang et al. (2022c)* indicate that the PRS derived from millions of COPD-related variants exhibits greater robustness compared to individual significant variants. In a study conducted by *Ma et al. (2020)*, the integration of Single Nucleotide Polymorphisms (SNPs) and clinical information demonstrated a robust predictive performance for COPD risk. Meanwhile, *Peng et al. (2020)* collected medical records, clinical features, and blood biochemical inflammation indicators. They analyzed and constructed multiple machine learning models, including the C5.0 decision tree, classification and regression trees, and iterative binary classification models. The research findings highlighted that the C5.0 decision tree classifier exhibited the best performance, allowing respiratory physicians to quickly assess the severity of COPD at an early stage (*Peng et al., 2020*). These studies exemplify the significant potential of PRSs and machine learning to improve COPD risk prediction and facilitate early intervention strategies for improved patient outcomes.

This study aims to develop and validate ML models to predict early-onset COPD (defined as COPD onset before 50 years of age) from PRS and patient data available in electronic medical records. We demonstrate the machine learning methods perform accurate prediction in general. In particular, models trained on personal information and blood biochemical results showed highest accuracy and sensitivity to predict COPD risks. As there is limited research on this topic, particularly in younger patients, this study will provide valuable information on COPD risk prediction and will help improve public health outcomes (*Saketkoo et al., 2014*).

# METHODS

## Data source

For this study, the UK Biobank was used as the data source (*Sudlow et al., 2015*). The sample was divided into a case group and a control group, the case group consisting of individuals with specific diseases, and the control group consisting of completely healthy individuals (*Hobbs et al., 2017*). This was done to more accurately compare the genomic data of the two groups to find the genes associated with the disease and also to facilitate the construction of machine learning-based classification models (*Li et al., 2019*). The data was

then divided into a training set and an independent test set. Considering the imbalance between positive and negative samples may have an impact on modeling and evaluation, we created ten sub-training sets. Each set consisted of the same positive samples and randomly sampled negative samples. The final model for one method was achieved by the best performance among these 10 models. Furthermore, to maintain a fair and unbiased comparison, we ensured the test set remained balanced (about 20% of all cases) (*Dietterich, 1998*). The data processing process can be seen in Fig. 1. In the final sub training set and independent testing set, both case and control samples were controlled between the ages of 20 and 50.

## Genome-wide association studies and polygenic risk score

Genome-Wide Association Studies (GWAS) was used to analyze the data (*Visscher et al., 2017*). GWAS is a research method used to study the correlations between genes and specific diseases, behavioral, or physiological traits (*Hindorff et al., 2009*). GWAS typically uses large samples of tens of millions of genetic loci (called single nucleotide polymorphisms, SNPs) analyzed to find SNPs associated with a disease or trait (*McCarthy et al., 2008*). In this study, plink was used for genotype and sample filtering, while a generalized linear model ('-glm' parameter) was used for GWAS research, with age and gender as covariates (*Purcell et al., 2007*). The data used here for statistics are performed on the complete training set. To predict the risk of the target disease, we employed the Polygenic Risk Score (PRS), a composite genetic index calculated based on the identified disease-associated SNPs (*Visscher et al., 2017*). We utilized significant correlation sites derived from GWAS analysis of the training set data. To compute individual PRS values, we assigned weights by summating the Odds Ratio (OR) values. Subsequently, we assessed and compared the predictive performance of PRSs against other modeling methods on the test set.

## Population filtering

Population filtering was used to remove samples with incorrect gender labeling, high SNP heterozygosity (PCA corrected heterozygosity > 0.1951), high feature missing rate (missingness > 0.02), and high genetic relationship correlation $\geq$ 0.05 (*Jian, Boerwinkle & Liu, 2014*). To ensure a single genetic background in the study, and to retain as many study individuals as possible, Anglo Whites were retained for analysis (*Bycroft et al., 2018*). Additionally, SNPs with an allele frequency > 0.01, deletion rate > 0.05, and Ha-Wen equilibrium test significance > 0.0001 were retained (*Wigginton, Cutler & Abecasis, 2005*).

## Machine learning methods

We used the R language to perform the entire data cleaning and modeling process. Drawing upon prior research, we utilized a diverse range of commonly employed machine learning techniques for feature modeling (*Alpaydin, 2014*; *Peng et al., 2020*; *Zhang et al., 2022a*), including neural networks (*Ripley, 2013*), Support Vector Machine (*Karatzoglou et al., 2004*) (SVM), Bayesian Generalized Linear Model (*Gelman & Hill, 2019*), Random Forest (*Wright & Ziegler, 2017*), C50 decision tree (*Kuhn, Weston & Coulter, 2018*),

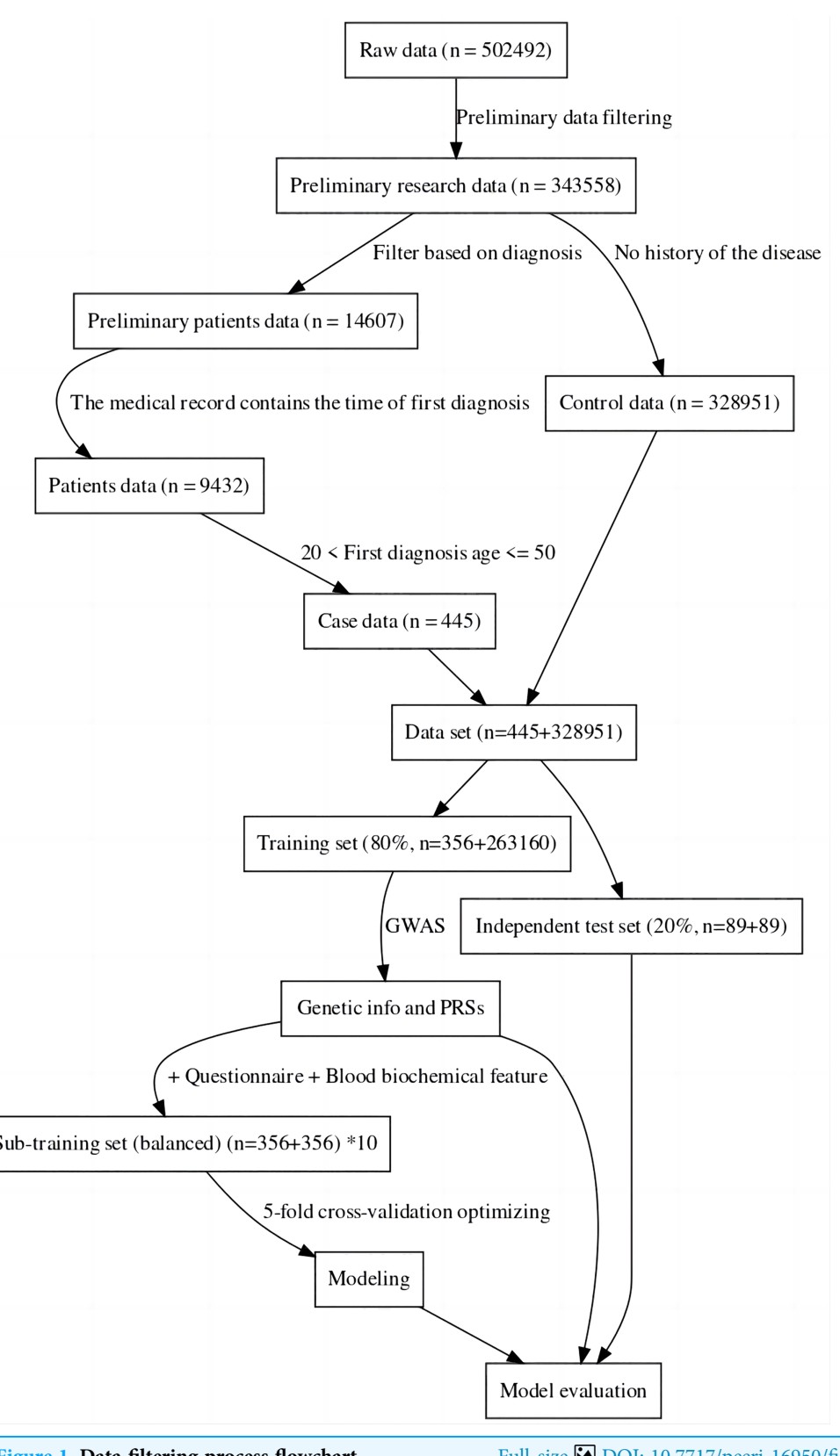

**Figure 1 Data filtering process flowchart.**

k-nearest neighbor (KNN) (*Kuhn, 2008*), AdaBoost (*Chatterjee, 2016*), and xgboost (*Chen, He & Benesty, 2016*). To ensure robust evaluation and control over model performance, we exclusively implemented five-fold cross-validation in the sub-training set (*Kuhn, 2008*). The parameter optimization for relevant models was facilitated using the train tool from the caret package (default optimization parameters) (*Kuhn, 2008*). Finally, we integrated multiple sub-training set models to evaluate the final independent test set.

## Feature and standards

The input features consisted of three types: genetic data, patient personal information data, and blood biochemistry data. The genetic data were derived from GWAS-derived SNP loci of significant association (*Visscher et al., 2017*). The personal information data of the patient contained basic information, diet, work, exercise, and alcohol and tobacco habits (*Garcia-Gil et al., 2014*). The blood biochemical data included the first blood biochemical index when patient information was entered in the UKB information base (*Bycroft et al., 2018*). The types of features used, as well as a brief definition and basic statistical summary of these features/variables, can be found in Table S1. For continuous data, date information was transformed into age-related integer values, while other integer data and numerical figures remained the same (*Wu et al., 2013*). In the case of categorical data with rank information, such data was converted into integer-type data based on the specified rank. Data consisting of only two factors were converted into binary integer-type data and multi-factor data kept factor data (*Hastie, Tibshirani & Friedman, 2009*). All numerical data were normalized during modeling.

## Performance evaluation

The choice of different classification thresholds can significantly impact outcomes. To ensure robust and comparable indicators, we extensively evaluated performance and compared models using the area under the curve (AUC), a comprehensive measure (*Hanley & McNeil, 1982*). Additionally, sensitivity and specificity metrics were computed to delineate specific model performance (*Akobeng, 2007*).

In our analysis, AUC serves as a global indicator of model performance across thresholds. It represents the probability of the model correctly ranking a random positive instance higher than a random negative instance.

To compute AUC, we employed receiver operating characteristic (ROC) curves, plotting sensitivity against 1-specificity across various thresholds. The AUC represents the area under this curve, offering an aggregate performance measure across all classification thresholds.

Sensitivity (Sensitivity) characterizes the model's ability to accurately identify positive cases. It is calculated as the ratio of true positives to the sum of true positives and false negatives.

$$\text{Sensitivity} = \frac{\text{True Positives}}{\text{True Positives} + \text{False Negatives}}$$

Specificity (Specificity) signifies the model's precision in recognizing negative instances, derived from the ratio of true negatives to the sum of true negatives and false positives.

$$\text{Specificity} = \frac{\text{True Negatives}}{\text{True Negatives} + \text{False Positives}}$$

For transparency and reproducibility, all code and results for performance testing and model comparison are publicly available at https://github.com/AilurusBio/ukb_analysis.

## RESULTS

The results of this study have revealed that machine learning can be used to predict the risk of developing early-onset COPD in individuals aged 20 to 50. The initial dataset contained 343,558 alternative samples from the UK Biobank raw data, from which a case-control group was created. Quality control measures were applied to both the samples and SNP loci to ensure the reliability of the study's findings. As shown in Fig. 1, after filtering for genotype quality control, the experiment utilized 784,256 SNP loci, with 429,288 remaining for analysis. The original dataset comprised 502,492 samples, out of which only a small number of individuals were diagnosed with COPD in their early years. Specifically, 14,607 individuals were diagnosed with COPD, as demonstrated by the age distribution shown in Fig. 2. To uphold the study's reliability, a meticulous selection process was undertaken, identifying 445 early-onset cases and 328,951 controls from the available samples. For independent testing, an equitable ratio of positive and negative samples, totaling 178 instances, was maintained at a 1:1 ratio. This approach ensured a fair and impartial assessment of the machine learning algorithm's performance.

GWAS analysis provided insight into the regions/loci associated with COPD in the genome. A total of 11 significant loci with a *p*-value of less than 1e−05 were identified and visualized using a Manhattan plot (Fig. 3). Additionally, the annotation of SNP allowed further analysis and identification of the degree of association of individual SNPs, with the most significantly correlated located at rs2158937 of AC005176.2 gene (Table 1).

The classification performance of PRS models that uses only genetic characteristics and weighting coefficients using OR values was evaluated in a test set. The results showed an accuracy of 0.52061, a sensitivity of 0.5118 and a specificity of 0.5294, which is insufficient for discrimination. However, better predictive ability was observed when data supplemented with blood biochemistry and questionnaire features was used in machine learning models. Table 2 shows the AUC performance of different machine learning models across different types of features. It can be seen that the random forest model has excellent predictive performance in various feature combinations in this task. The random forest model has a best AUC performance of 0.8505, a sensitivity of 0.7584 and a specificity of 0.8539 (Fig. 4, Table 2). The sensitivity, specificity, and balanced accuracy indicators of all models can be found in Table S2. This suggests that combining personal patient information and biochemistry characteristics improves the efficacy of machine learning models to predict the risk of early-onset chronic obstructive pulmonary disease.

Furthermore, to reveal the metrics that are mostly contributing the COPD risk, we analyzed the importance of input features for the optimal model, and revealed that

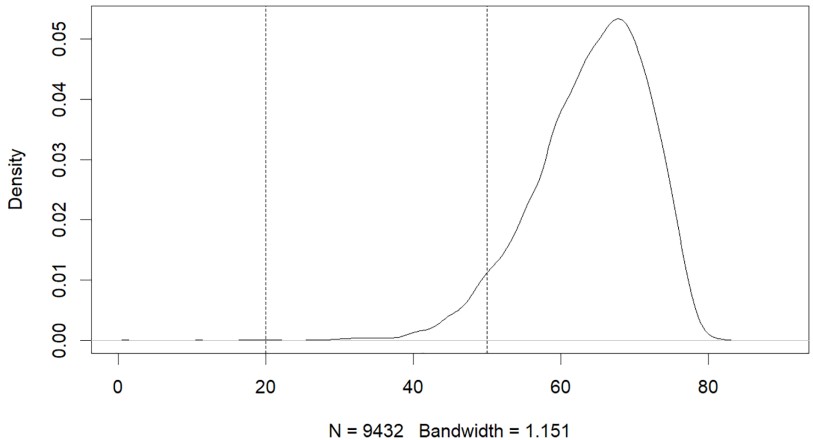

**Figure 2 Age distribution of patients with COPD.**

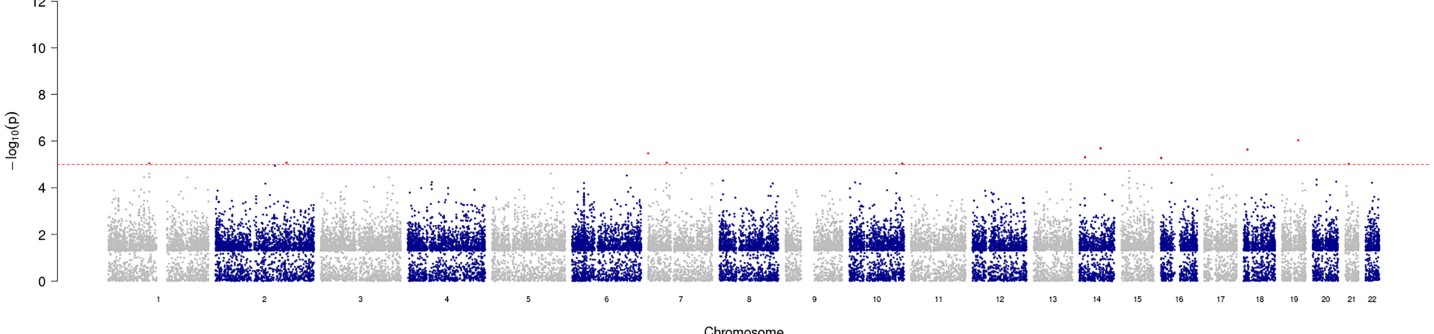

**Figure 3 Manhattan plot of significant SNPs across the genome.**

**Table 1 Variant information.** The table displays the chromosome number (CHROM), position (POS), identifier (ID), reference (REF) and alternate (ALT) alleles of each variant, as well as the odds ratio (OR), *P*-value (P), gene name (GENE NAME), and distance to the nearest gene (DISTANCE).

| CHROM | POS | ID | REF | ALT | OR | *P* | GENE_NAME | DISTANCE |
|---|---|---|---|---|---|---|---|---|
| 19 | 40,132,472 | rs2158937 | C | T | 1.88713 | 9.0e−07 | AC005176.2 | 0 |
| 14 | 72,769,920 | rs117757342 | A | G | 2.82681 | 2.0e−06 | RGS6 | 0 |
| 18 | 9,406,015 | rs79126399 | A | G | 2.63550 | 2.3e−06 | TWSG1 | 3,596 |
| 7 | 522,043 | rs118051507 | G | A | 2.79461 | 3.4e−06 | AC147651.1 | 1,700 |
| 14 | 34,755,751 | rs79434931 | C | T | 2.76979 | 5.0e−06 | EGLN3 | 0 |
| 16 | 1,200,232 | rs62012311 | C | T | 2.22736 | 5.3e−06 | CACNA1H | 3,008 |
| 2 | 175,349,076 | rs115361661 | C | T | 2.99481 | 8.4e−06 | GPR155 | 0 |
| 7 | 46,167,359 | rs73107888 | G | A | 2.76816 | 8.5e−06 | AC023669.2 | 17,175 |
| 1 | 102,471,839 | rs1415094 | T | C | 1.56340 | 9.0e−06 | OLFM3 | 9,252 |
| 10 | 131,080,127 | rs11016735 | A | G | 2.27625 | 9.3e−06 | RP11-168C9.1 | 0 |
| 21 | 23,757,543 | rs35441718 | C | T | 1.66449 | 9.5e−06 | AP000705.7 | 0 |

**Table 2 Analysis of AUC performance of different machine learning models.** Columns represent different machine learning models, including artificial neural network (ANN), support vector machine, k-nearest neighbors (KNN), Random Forest (RF), AdaBoost, generalized linear model with Bayesian regularization (Bayes.glm), xgBoost and the C5.0 decision tree algorithm. Rows represent different types of characteristic information, including genetic information obtained by GWAS (G), personal information obtained by questionnaires (Q), and blood biochemical information (B). The values in the table represent the AUC results of each model under the corresponding features, where a higher AUC indicates better classification performance.

| Features | ANN | SVM.Linear | SVM.Radial | KNN | RandomForest | AdaBoost | Bayes.glm | C5.0 | xgBoost | SVM.Poly |
|---|---|---|---|---|---|---|---|---|---|---|
| G | 0.54684 | 0.54646 | 0.54659 | 0.56577 | 0.57045 | 0.55252 | 0.54431 | 0.53649 | 0.53611 | 0.54772 |
| Q | 0.7946 | 0.82212 | 0.84156 | 0.67542 | 0.848 | 0.82389 | 0.83045 | 0.83695 | 0.8384 | 0.84459 |
| Q + B | 0.77061 | 0.75319 | 0.82187 | 0.50619 | 0.85052 | 0.83626 | 0.80141 | 0.80457 | 0.82793 | 0.83411 |
| G + Q + B | 0.72327 | 0.76038 | 0.81113 | 0.50619 | 0.84951 | 0.82565 | 0.79447 | 0.8254 | 0.83171 | 0.823 |

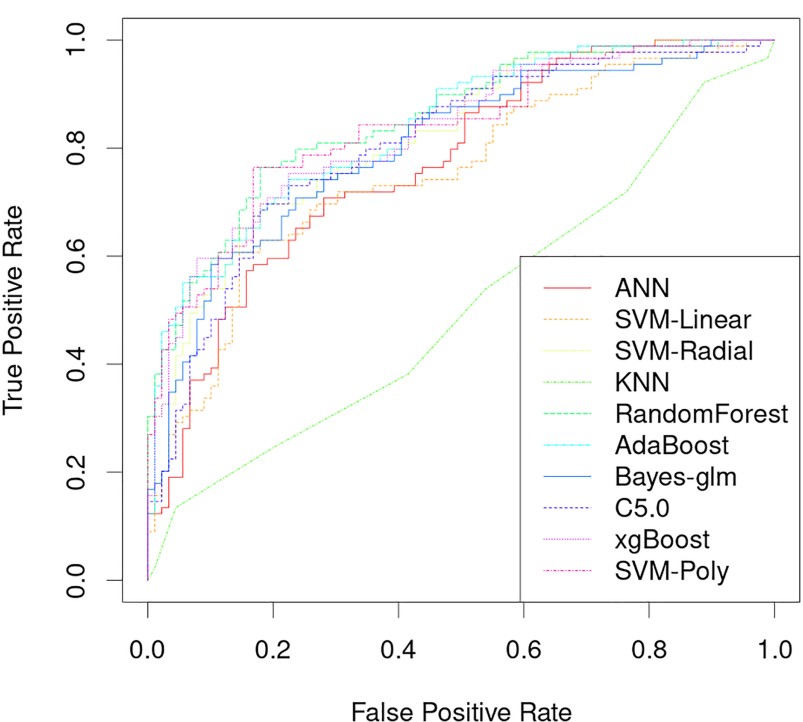

**Figure 4 Performance analysis of different machine learning models on receiver operating curve (ROC).**

smoking status was strongly correlated with the prediction of COPD, as well as Townsend deprivation index at recruitment and current tobacco smoking (Fig. 5).

## DISCUSSION

The results of this study demonstrate that machine learning models can be used to accurately predict the risk of developing early-onset COPD at age 20 to 50 years. The best model achieved an AUC of 0.8505 and a specificity of 0.8539 (Fig. 4), which is significantly better than previous studies that use only genetic characteristics for prediction (*Singh et al.,*

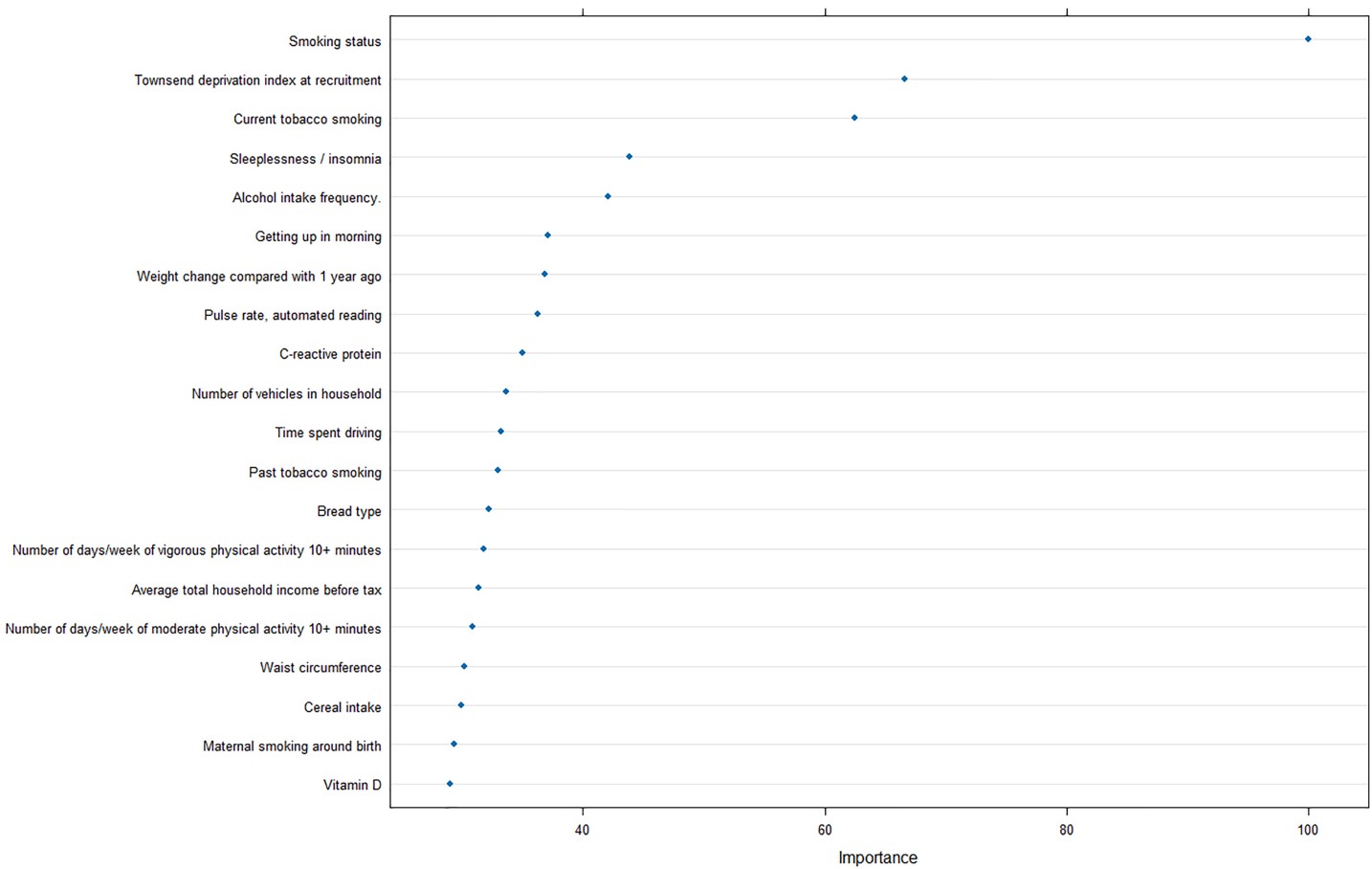

**Figure 5 Feature importance analysis in optimal machine learning model.**

*2019*). It should be noted that in this study, random forest has good performance in multiple predictions, but this does not mean that the random forest model is the optimal model for screening the early risk of the disease. Multiple machine learning models have similar predictive performance, and these weak performance differences may be caused by methods of adjusting hyperparameters or the quality of the dataset itself.

In addition, the importance analysis of the optimal model can provide interpretable features regarding the COPD risk, smoking, and socioeconomic status. Smoking is the most common cause of COPD and the harmful chemicals in tobacco smoke can cause inflammation and damage to the airways and lungs, leading to the development of COPD over time (*Mathers & Loncar, 2006*). Furthermore, exposure to other pollutants such as biomass fuel and air pollution has also been linked to an increased risk of COPD (*Fullerton, Gordon & Calverley, 2009*).

The finding that socioeconomic deprivation is also a significant risk factor for COPD is concerning, as it suggests that individuals from lower socioeconomic backgrounds may be at a greater risk of developing COPD due to factors such as increased exposure to pollution, poor access to healthcare, and limited opportunities for education and employment. Quitting smoking, reducing exposure to pollutants, and improving access to

healthcare and education can help reduce the incidence and burden of COPD in high-risk populations.

Despite the promising results of this study, there are several limitations that should be noted. First, the data used in this study was from a single population, and the results may not be generalizable to other populations. Second, the data used in this study was retrospective, so the predictive performance of the model may be affected by recall bias.

## CONCLUSIONS

In conclusion, this study demonstrates that machine learning models can be used to accurately predict the risk of developing early-onset COPD in individuals aged 20 to 50 years. Our model achieved an AUC of 0.8505 and a specificity of 0.8539, indicating that it is capable of accurately predicting COPD risk. The model was further improved when personal information and blood biochemical test data were included in the analysis. These findings suggest that ML models can be used to accurately predict COPD risk in individuals aged 20 to 50 years, and may provide a valuable tool for early detection and intervention. Further research is needed to investigate the potential of these models to improve the accuracy of COPD risk prediction in other populations and to identify additional factors that may influence the development of COPD.

## ACKNOWLEDGEMENTS

Heartfelt appreciation is expressed to the family members of the research team for their unwavering encouragement and assistance. Their patience and understanding have been essential to the completion of this work. We would like to express our gratitude to the UK Biobank for providing access to their valuable data resources, which enabled the successful completion of this research project.

### Funding

This work has received funding and technical support from the Ailurus Biotechnology Co., Ltd. The funders had no role in study design, data collection and analysis, decision to publish, or preparation of the manuscript.

### Grant Disclosures

The following grant information was disclosed by the authors:
Ailurus Biotechnology Co., Ltd.

### Competing Interests

Jiani Hu, Jianzhe Yang, and Jie Song are currently employed by Ailurus Biotechnology Ltd. and have contributed to the research and development of the materials and methods described in this article. Therefore, they may have a financial interest in the outcome of this study. However, this does not alter our adherence to the journal's policies on data sharing and integrity.

## Author Contributions

- Guanglei Liu conceived and designed the experiments, performed the experiments, analyzed the data, prepared figures and/or tables, authored or reviewed drafts of the article, and approved the final draft.
- Jiani Hu performed the experiments, prepared figures and/or tables, and approved the final draft.
- Jianzhe Yang performed the experiments, authored or reviewed drafts of the article, and approved the final draft.
- Jie Song performed the experiments, analyzed the data, prepared figures and/or tables, authored or reviewed drafts of the article, and approved the final draft.

## Data Availability

The data and code (preprocessed analysis data, model results, and relevant code files) are available at GitHub and Zenodo:

- https://github.com/AilurusBio/ukb_analysis.

- Ailurus Biotechnology Co., Ltd. (2024). Dataset from: Predicting early-onset COPD risk in adults aged 20–50 using electronic health records and machine learning. (Version v1). Zenodo. https://doi.org/10.5281/zenodo.10491364.

## Supplemental Information

Supplemental information for this article can be found online at http://dx.doi.org/10.7717/peerj.16950#supplemental-information.

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
