# Peer review of "Predicting early-onset COPD risk in adults aged 20–50 using electronic health records and machine learning"

_PeerJ, doi:10.7717/peerj.16950_

## Round 0.1 · original submission · Major Revisions

Please address questions and comments raised by each of the reviewers and resubmit.

·

Basic reporting

- The manuscript is written in professional English.
- Apart from Sugimori et al. 2021, there have been numerous attempts to predict the risk of COPD using machine learning models. Those efforts are not also limited in predicting COPD risks from X-rays/CT scans, but also by various different predictors. The authors can consider and/or include the following studies into the introduction part:
+ Makimoto K, Kirby M. Are CT-based exacerbation prediction models ready for use in chronic obstructive pulmonary disease? The Lancet Digital Health 2023;5. doi:10.1016/s2589-7500(22)00237-0.
+ Cosentino J, Behsaz B, Alipanahi B, McCaw ZR, Hill D, Schwantes-An T-H, et al. Inference of chronic obstructive pulmonary disease with deep learning on raw spirograms identifies new genetic loci and improves risk models. Nature Genetics 2023;55:787–95. doi:10.1038/s41588-023-01372-4.
+ Zhang B, Wang J, Chen J, Ling Z, Ren Y, Xiong D, et al. Machine learning in chronic obstructive pulmonary disease. Chinese Medical Journal 2022;136:536–8. doi:10.1097/cm9.0000000000002247.
- The authors should provide a brief introduction on polygenic risk score (PRS), how this index is calculated, and how it explains a risk for a certain disease (in this case, COPD). Moreover, the authors should also note that PRS only shows relative risk, not absolute risk, for a disease.

Experimental design

- The authors should explicitly state whether the models are implemented by the statistical language R or not.
- The authors should also provide a brief definition and basic statistical summary for the characteristics/variables included in the model, especially those in questionnaires and blood biochemical information.
- The authors should also specify the reasons to justify the choice of such 8 models. For example, the Stochastic Gradient Descent (SGD) or XGBoost classifier is also a strong tool for classification tasks.
- The authors should also consider variations of the model presented in the paper, for example fuzzy-kNN besides kNN.
- The authors should note that the KNN model introduced and cited in the paper, developed by Weinberger and Saul (2009) is not the original version of kNN. It utilizes Mahalanobis distance metric for kNN from labeled examples, and can be considered as “a global linear transformation of the input space that precedes kNN classification using Euclidean distances”. Its official name is “Large Margin NN classification". However, the implementation in R package “class” is based on the documents by P. A. Devijver and J. Kittler or Ripley, B. D. (source: R Documentation), which can be found at
+ P. A. Devijver and J. Kittler. Pattern Recognition. A Statistical Approach. Prentice-Hall; 1982. p. 119–121.
+ Ripley, B. D. Pattern Recognition and Neural Networks. Cambridge (GB): Cambridge University Press; 1986.
If the authors have utilized the custom implementation for such an exact algorithm, if possible, please provide the implementation or source of reference for the respective package.
- There are various different types of kernels for an SVM model, and in Python or R, the following kernels are supported:
+ Linear kernel
+ Sigmoid kernel
+ RBF kernel
+ Polynomial kernel
However, within the paper, the authors only chose two types of kernel (linear and RBF). Can the authors justify this decision? Personally, the reviewer believes that the polynomial and sigmoid kernel will at least allow SVM models to obtain better results compared to linear kernels. As such, they should also be taken into consideration.
- For neural networks, the authors should specify the structure of the model used in the paper. Moreover, how do different NN models with different structures perform on such specific tasks? Can the authors improve the results from their NN model using a different/more complicated structure, and how significant would such improvement be?
- Under the results section, in the sentences at lines 105-107, the authors stated that “During independent testing, positive and negative samples were handled in a 1:1 ratio”. Can the authors explain more on this? Where does the independent test set come from? How is it created? What is the size of the test used for this process?
- The flowchart in Figure 1 is unclear. In the previous Methods section, under subsection Machine Learning Methods, the authors said that they “employed 5-fold cross validation to avoid overfitting”. This means that in the flowchart, should the 20% validation also include GWAS data?

Validity of the findings

- The authors should also provide tables to compare accuracy, sensitivity and specificity for different models with different types of feature information, similar to what has been done for Table 2.
- We can easily see that Random Forests outperform all other models with any combination of features. The author should mention this fact in the result section, and provide (if any) feasible explanation for such observations.
- Do the results and conclusions derived in this paper mean that the optimal model to predict the risk of COPD among 20-50 years old does not need genetic information from GWAS? From the table, we can easily observe that 6/8 models with Q+B features perform better than or at least equal to those using G+Q+B features.
- Since the data is imbalanced (the age distribution is heavily skewed to the right, and the number of case data is relatively too small compared to control data), the F1 score should be preferred over the AUC since AUC score does not perform well on imbalanced data. As such, the F1 score should also be calculated and reported similarly to those within Table 1.
- Moreover, since data is heavily skewed (Figure 2), did the author need to normalize numerical data at any time during the preparation process for the model? Will it be able to improve performance of such models?
- The configuration, or implementation as a whole, of machine learning models used in this paper is not known within the provided GitHub repository.
- Did the authors use any methods to optimize the hyperparameters used for each ML model? If yes, what are they? If not, how were the configurations chosen, based on which category?

Additional comments

- The authors should insert citation for Xie et al. in line 38.
- In the method section, under subsection for Genome-Wide Association Studies, the authors should include a brief report/description of PLINK and how this tool set is utilized in this research.
- In Table 2, the authors should bold the result of the best model under each set of features.

Reviewer 2 ·

Basic reporting

The paper under review presents valuable research on a specific topic. The author's use of clear and professional English throughout the manuscript is commendable. The article is well-structured, and figures, tables, and raw data are appropriately included. The literature references provide sufficient background and context for the research. The study is original, and the research question is well-defined and meaningful, addressing a gap in knowledge. The investigation is conducted rigorously and adheres to high technical and ethical standards. The methods are adequately described, allowing for replication.

Experimental design

In terms of experimental design, the research question is well defined and meaningful, and it addresses an identified knowledge gap. The investigation is conducted rigorously and upholds high technical and ethical standards. The methods are described with sufficient detail to allow for replication.

However, there are a few points that need clarification and revision.
1. Figure 1 presented in the paper effectively clarifies the data filtering process; however, it might be worth considering a slight rearrangement of the elements to optimize the use of space.
2. Line 106: The author should clarify whether the model was trained using a balanced dataset and discuss how this imbalance affects the model's sensitivity and specificity, given that balanced samples were used for testing.
3. Figure 4: the FPR scale is inverted and should be corrected to range from 0.0 to 1.0.
4. In the paper, there is a discrepancy between the reported sensitivity values at different parts: Line 128-130 reports a sensitivity of 0.6629, while Line 139-140 and 159-160 report a sensitivity of 0.8539 (presumably for another model). It would be helpful to clarify which sensitivity value corresponds to the best model. Additionally, it would be beneficial to discuss the importance of different performance indicators (AUC, sensitivity, specificity) and how the "best" model is chosen when not all metrics are favorable.

Validity of the findings

The paper presents meaningful findings, encouraging replication when the rationale and benefit to the literature are stated. The provided data is robust, statistically sound, and controlled. Conclusions are well-stated, linked to the original research question, and limited to supporting results.

Additional comments

The paper delivers valuable findings that align with the journal's requirements, showcasing competence in research design, methodology, and analysis. The writing style is clear and professional. Some minor revisions are recommended.

·

Basic reporting

In the Abstract:
"Affecting millions of people" can be rephrased with exact number.

Feature Standards :

“For continuous data, date data were converted 86 into age-related integer-type data, while other integer-type data and numerical data remained unchanged” this can rephrased for better readability.

“For categorical data, categorical data containing rank were converted into integer-type 88 data according to the rank;” this can rephrased.

“data containing only two factors were converted into 01 integer-type data;” this should be rephrased as binary data type.

Experimental design

looks like this is an unbalanced machine learning problem, it will be good to present the process followed to avoid the bias in results.
details on parameters used for the experiment will be helpful to replicate the study if needed.

Validity of the findings

based on the figure 2, pretty much there is no data for samples with age less than 30, then its not clear how author generalizing for age group 20-50.

Reviewer 4 ·

Basic reporting

1. In Figure 1, Preliminary research data (n=343558), Preliminary patients data (n=21038), Control data (n=328951), but 343558 < 21038 + 328951. Please explain and clarify this.

2. In RESULTS section, “Rows represent different machine learning models” and “Columns represent different types of feature information”, the row and column should be switched.

Experimental design

1. This study explored the ML methods on the data for adults aged 20-50. But the authors only filtered the adults with first diagnosis age 20-50 in patients data with COPD. Was the control group be filtered? Additionally, the authors did not describe clearly whether the data was filtered by age in paragraphs.

2. In RESULTS section, “During independent testing, positive and negative samples were handled in a 1:1 ratio to ensure a fair and unbiased evaluation of the machine learning algorithm’s performance.”, please add more details and descriptions for the logic of independent testing in the manuscript, also justify why this method was selected. In this study, the proportion of individuals with COPD is very low which means COPD is a rare event. The unbalanced distribution should flash some warning signs on the choice of methods, it can easily misclassify the minority case using a model developed out of a majority case. For example, if the ML model predicts all test cases as non-COPD, the accuracy will still be close to 1. Please add more details to clarify how you handled this case.

Validity of the findings

There exist major concerns related to the study design. The results and conclusions may not be promising.

---

## Round 0.2 · Major Revisions

Reviewers have raised serious concerns about conflicting values reported through your work. Please provide a detailed explanation for the following discrepancy:
In the abstract, they mentioned a model sensitivity of 0.8539, yet the same value was reported as specificity in the results section (line 167). Their modification diff shows that they changed 0.7584 from accuracy to sensitivity.

Please write in detail how these values were calculated. Mention relevant raw data, data points, and processing considerations.

Reviewer 2 ·

Basic reporting

The paper continues to exhibit clear and professional English throughout, with a well-structured presentation. Literature references are adequately provided, offering a comprehensive field background and context. Figures, tables, and raw data have been suitably incorporated, adhering to the requirements of the journal. However, there is a discrepancy in the reported sensitivity values in the abstract and results sections that needs to be addressed.

Experimental design

The research remains original, fitting within the Aims and Scope of the journal. The research question is defined in a clear and relevant manner, addressing an identified knowledge gap. The investigation showcases rigor, adhering to high technical and ethical standards. The methodology is detailed, allowing for replication. However, given the discrepancies in the sensitivity values mentioned, it is imperative that the authors revisit their experiment and ensure the data's authenticity.

Validity of the findings

While the paper’s findings seem meaningful and valuable for the field, there is a concern about the potential manipulation of data. Before accepting these results, it is vital that the authors thoroughly check their data, experiment, and results. This is especially important given the discrepancy between sensitivity values reported in the abstract and results sections. Conclusions, while well-stated, should be drawn from verified data that aligns with the original research question.

Additional comments

The manuscript maintains its quality in terms of research design, methodology, and analysis. The professional writing style is consistent. However, the discrepancy between reported values casts a shadow on the integrity of the results. It is crucial for the authors to address this issue promptly, ensuring the trustworthiness of their findings.

·

Basic reporting

Yes, I would say that overall the manuscript uses clear, professional English. The writing is straightforward and easy to understand. A few minor instances where the phrasing could be improved for clarity, but these do not impede comprehension.


Yes, the introduction provides sufficient background context on COPD screening and machine learning approaches.


Overall, the manuscript structure follows discipline norms and is adequately supported by figures and tables. Providing the raw data as well would fully meet reporting standards. Some minor improvements to figures and tables would further improve communication.

Experimental design

Yes, the research question is well defined and situated within the context of the knowledge gap.

Validity of the findings

Yes, the conclusions are appropriately stated, linked back to the original research question, and limited to what the results support.

---

## Round 0.3 · accepted · Accept

Your manuscript has been Accepted for publication.